# Instantaneous Ablation Behavior of Laminated CFRP by High-Power Continuous-Wave Laser Irradiation in Supersonic Wind Tunnel

**DOI:** 10.3390/ma16020790

**Published:** 2023-01-13

**Authors:** Te Ma, Jiangtao Wang, Hongwei Song, Ruixing Wang, Wu Yuan

**Affiliations:** 1Key Laboratory for Mechanics in Fluid Solid Coupling Systems, Institute of Mechanics, Chinese Academy of Sciences, Beijing 100190, China; 2School of Engineering Science, University of Chinese Academy of Sciences, Beijing 100049, China; 3State Key Laboratory of High-Temperature Gas Dynamics, Institute of Mechanics, Chinese Academy of Sciences, Beijing 100190, China

**Keywords:** laminated CFRP, laser irradiation, in situ observation, instantaneous ablation behavior, coupled thermal-fluid-ablation model

## Abstract

Experimental and numerical investigations of the instantaneous ablation behavior of laminated carbon fiber-reinforced polymer (CFRP) exposed to an intense continuous-wave (CW) laser in a supersonic wind tunnel are reported. We establish an in situ observation measurement in the experiments to examine the instantaneous ablation behavior. The surface recession depth is calculated by using the Particle Image Velocimetry (PIV) method, taking the ply angle of laminated CFRP as a reference. A coupled thermal-fluid-ablation numerical model incorporating mechanisms of oxidation, sublimation, and thermomechanical erosion is developed to solve the ablation-through problem of multilayer materials. The results show that the laser ablation depth is related to the laser power density, airflow velocity and airflow mode. Thermomechanical erosion is the primary ablation mechanism when the surface temperature is relatively low and the cavity flow mode is a closed cavity flow. When the surface temperature reaches the sublimation of carbon and the airflow mode is transformed to open cavity flow, sublimation plays a dominant role and the ablation rate of thermomechanical erosion gradually decreases.

## 1. Introduction

The laminated carbon fiber-reinforced polymer (CFRP) composites have been widely used in industrial sectors, aeronautics and aerospace engineering. The high specific strength, specific stiffness, and excellent designability characteristics of laminated CFRP can significantly improve the performance of high-speed flights such as missiles and unmanned aerial vehicles. However, the thermal damage caused by laser irradiation is a significant threat when the high-power laser becomes an important intercept strategy [1,2,3].

Under the combined action of laser irradiation and tangential airflow, complex ablation behaviors may occur, such as pyrolysis of the polymer matrix, oxidation and sublimation of residual char and fibers, and thermomechanical erosion. Several works have been reported on the laser ablation of polymer-based composites [4,5,6,7,8,9,10,11,12,13,14,15,16,17,18]. There are few studies considering the influence of tangential airflow. Recently, we presented the comparative experimental results of the ablation behavior of laminated CFRP in different environments, that is, static nitrogen, stagnant air, open airflow, and tangential supersonic airflow. The thermomechanical erosion (TME) caused by tangential airflow not only increases the total ablation depth at the coupled ablation zone (CAZ) but also converts the heat-affect zone (HAZ) into the downstream affected zone (DAZ) [19]. In our previous work, the traditional method is used to compare the experimental results of specimens, such as the linear ablation rate, before and after experiments [20,21,22,23,24,25]. However, the laser ablation process is complex, resulting from the mutual coupling of various ablation mechanisms. The polymer matrix of the CFRP composites undergoes a pyrolysis reaction first and generates the pyrolysis gas and residual char. Fibers that have lost matrix reinforcement are subject to mechanical erosion by tangential airflow. Then, fibers also undergo an oxidation reaction or even a sublimation reaction as the temperature rises. The biggest drawback of the traditional method is failed to provide real-time information about the laser ablation process. Thus, instantaneous data are demanded to analyze the evolution of ablation mechanisms. Some researchers develop real-time ablation methods such as ablation potentiometers, thermocouple arrays, and ultrasonic waves to acquire real-time ablation data. McWhorter et al. [26,27,28] developed an ablative potentiometer for real-time measurement of ablative characteristics. The thermocouple is embedded in the insulation layer to measure the dynamic ablative process of the head insulation layer. Cauty et al. [29] employed an ultrasonic method to assess the dynamic ablative process of the insulation layer in the engine. Sakai et al. [30] described an ablation sensor that can measure ablation fronts in real-time and in situ, but the measurement is limited to a small region. Martin et al. [31] devised a technique for obtaining real-time X-ray radiography of ablated materials. Qu et al. [32] obtained real-time ablation images using an optoelectronic system for in situ observation and measurement in the arc-heated tunnel test. Tang et al. [33] analyzed the thermal ablation mechanisms of C/SiC composites based on an in situ and real-time optical visualization technique. Zhu et al. [34] used an optical imaging technique to in situ and real-time record the surface evolution of a flat plate subjected to thermal ablation at 1700 °C in a wind tunnel. Nonetheless, the temperature of the specimen under laser irradiation is more extreme. For the indirect measurement method, the high temperature and the laser reflection cause the image to be overexposed, covering up the adequate information of the image. The above experimental methods are insufficient to obtain instantaneous ablative morphology under complex high-temperature environments. Therefore, it is necessary to establish an in situ observation technology that can be applied to the combined action of high-energy laser and wind tunnel to clarify the influences of tangential airflow on laser ablation behaviors.

Meanwhile, the numerical simulation has been a powerful tool to assist the interpretation of the experimental results and reveal the ablation mechanism [35,36]. However, laser irradiation is a particular local ablation and fluid–solid interface recession problem. The traditional method uses Arbitrary LagrangianEulerian (ALE) adaptive remesh algorithm to adjust the location of nodes [37]. Nevertheless, it is unsuitable for laminated CFRP composite (the different material orientations can be performed as one material) due to the ALE method being applicable to single-layer material [38,39]. Furthermore, the coupled thermal-fluid-ablation behaviors become more complex under the combined action of laser irradiation and tangential airflow. For instance, we discovered a unique “avalanche” phenomenon in which the ablation rate under tangential airflow was significantly higher than that in the static air environment for C/SiC composites [40]. The results show that laser ablation has complex coupling behaviors in tangential airflow. The evolution of the laser ablation pit changed the local airflow mode, and the unsteady change of the local aerodynamic force significantly altered the laser ablation behavior. Therefore, a comprehensive coupled thermal-fluid-ablation model that deals with the multilayered ablation-through problem should be established to identify the laser ablation mechanism and analyze the coupling behaviors.

This paper aims to obtain experimentally and numerically the instantaneous ablation behaviors of laminated CFRP. The contribution of various ablation mechanisms to total ablation is revealed. Firstly, an experimental method is developed to measure the instantaneous ablative morphology evolution of the laser irradiation surface of laminated CFRP composites in a supersonic wind tunnel. The experimental method includes an in situ optoelectronic measuring system appropriate for extreme thermal–mechanical conditions. In addition, Particle Image Velocimetry (PIV) is used as a post-processing method to calculate the surface recession depth (SRD). Then, a coupled thermal-fluid-ablation model is developed to demonstrate the contribution of each ablative mechanism. The Radial Basis Function (RBF) and Remapping Solution Technology (RST) methods are established to simulate the multilayer ablation-through behaviors. The evolution of ablative mechanisms and morphology of multilayered CFRP, when subjected to continuous-wave (CW) laser irradiation and supersonic airflow, are revealed.

## 2. Experimental Setup

### 2.1. Experimental Procedure

The experimental setup is illustrated in Figure 1. An in situ observation measurement consists of a high-speed camera, neutral filter, and narrow-band pass filter. Laser illumination provides background light while avoiding high-energy laser bands with a narrow-band pass filter. The laser device, laser illumination, and high-speed camera are triggered and started simultaneously by a synchroscope during the experiments to ensure sampling consistency.

The laminated CFRP specimen is placed in the test chamber of a supersonic wind tunnel using a specially designed fixture that ensures the tangential flow of the supersonic free stream over one side of the specimen. During the test, this side is exposed to laser irradiation from a Yb_YLS continuous-wave laser with a wavelength of 1064 nm. The laser beam irradiates the CFRP sample through a high-temperature quartz window. The experiment employs the supersonic wind tunnel facility at the State Key Laboratory of High-Temperature Gas Dynamics (LHD) of the Institute of Mechanics, Chinese Academy of Sciences. It operates on the oxygen-hydrogen combustion principle and can provide a free stream of Mach 1.8 to 4.0 in the test section. The principle of the wind tunnel can also be seen in [19]. In the experiments, tangential supersonic airflow is set as Mach 1.8 and Mach 3.0, respectively. The diameter of the laser spot irradiated on the sample is 10 mm. The laser irradiation time is set as 4.0 s. Two groups of laser parameters, that is, power densities of 1273 W/cm^2^ and 2546 W/cm^2^, are tested. 

Laminated T700/BA9916 CFRP plates with the ply pattern of [45°/0°/−45°/90°]_2S_ are fabricated and provided by AVIC Composite Corporation Ltd. (Beijing, China). Moreover, BA9916 is a 180 °C high-temperature curing and high-toughness epoxy resin matrix. The detailed preparation process of the specimen can be found in [19]. Table 1 shows the main physical and mechanical properties of the T700/BA9916 CFRP plates. The samples used in the experiment have dimensions of 50 × 50 × 2.40 mm and a lamina thickness of 0.15 mm.

### 2.2. Method of Image Analysis

We use Particle Image Velocimetry (PIV) methods to calculate the fibers’ velocity distribution on the laser irradiation surface of the laminated CFRP. In the present paper, the fibers serve as the tracer particles for measuring velocity. As illustrated in Section 2.1, the experimental procedure does not include a particle pattern typically used to visualize fluid motion (either gaseous or liquid). The free, open-source code PIVlab is employed [41], together with a Matlab code that is based on a state-of-the-art multi-pass window deformation algorithm. PIVlab is reported to provide relatively accurate measurements with a random error, typically less than 0.02 pixels/frame, under ideal synthetic images with no noise and no shear. Figure 2 depicts the flowchart of PIVlab 2.02 software.

After determining and analyzing the size of the ablation pit, the captured images are converted to greyscale images and imported into PIVlab in Matlab R2016a software. An interrogation area of about 5×5 mm of the two-dimensional plate is chosen as the region of interest (ROI) to perform the analysis. According to a general recommendation, three passes are used for data analysis, 128 pixels in Pass 1, 64 pixels in Pass 2 and 32 pixels in Pass 3. Inhomogeneous lighting caused by laser irradiation can result in low-frequency background information. It can be eliminated by a high-pass filter that mostly conserves the high-frequency information from particle illumination. The Fast Fourier Transform (FFT) is used to solve the problem in the frequency domain.

## 3. Coupled Thermal-Fluid-Ablation Model

In our previous work, we preliminarily analyzed the thermal-fluid–solid coupling behaviors [19]. The numerical simulation includes a thermomechanical model, linear ablation models, and the heat balance analysis on the ablation surface to determine the ablation depth and clarify the ablation mechanism. Meanwhile, Thermal Gravimetry (TG) and Differential Scanning Calorimetry (DSC) tests are performed to obtain the pyrolysis process of epoxy resin, the oxidation process of carbon fiber, and the reaction heat.

Nevertheless, the previous work equates the laminated CFRP to a macroscopic model and cannot reveal the ablation process through each layer. Each layer has a different ply angle for the multilayer laminated CFRP composite. Therefore, each layer can be treated as a different material, and a more precise model is demanded to examine the ablation behavior. Furthermore, there needs to be more analysis of the relationship between thermomechanical erosion and tangential airflow. In order to solve the above two deficiencies, we further improve the numerical model of the previous work focused on the coupled ablation zone (CAZ).

First, the coupled numerical analysis and the theoretical model are introduced. Figure 3 depicts the flowchart of the coupled thermal-fluid-ablation analysis. The entire analysis consists of three steps. Step 1 is material properties at high temperatures, where a pyrolysis model of the epoxy resin matrix and mass balance are used. Step 2 is thermochemical ablation and thermomechanical erosion analysis, where a linear ablation model and heat balance are used. Step 3 is multi-physics coupling analysis, where the interactive and coupled behavior of ablation, fluid and structure are investigated.

The fluid and structural solver solve the partition problem independently, and coupling analysis is realized by controlling the data transfer on the fluid–solid boundary. The coupling strategy employs a loosely coupled partitioned strategy to reduce computational efforts by exchanging information less frequently between solvers. Step 1 provides multi-scale thermomechanical properties for Step 2 in each time step to ensure the correctness of HAZ. A multi-scale analysis model to identify the high-temperature parameters can be identified in [42]. The material content change model can be expressed using Arrhenius functions [19].
(1)$$\frac{\partial \varphi_{b}}{\partial t}=-J_{b}\varphi_{b}^{n}exp\left(-\frac{E_{Ab}}{RT_{w}}\right)$$
(2)$$\frac{\partial \varphi_{f}}{\partial t}=-J_{f}{\left[\varphi_{f}-\varphi_{f}^{0}\left(1-\Gamma_{f}\right)\right]}^{n}exp\left(-\frac{E_{Af}}{RT_{w}}\right)$$
where *φ* is the concentration, subscripts *f* and *b* represent fiber and matrix, respectively, *E_A_* denotes the apparent energy of activation, *J* is the pre-exponential factor, *t* is the time, *Γ_f_* is the gasification coefficient of fiber, *R* is the universal gas constant, and *T_w_* is the temperature, *n* is the reaction order.

In Step 2, the fluid field transfers the aerodynamic heat flow and pressure to the structure field via the fluid–solid interface (FSI). The following equations calculate the boundary movement of each node caused by linear ablation [19]:(3)$$v_{t}=v_{o}+v_{s}+v_{m}$$
(4)$$v_{o}=\frac{1}{\rho_{w}}\sum \frac{M_{i}}{M_{o_{2}}}p_{o_{2}w}^{n}A_{i}exp\left(-\frac{E_{Ai}}{RT_{w}}\right)$$
(5)$$v_{s}=\frac{1}{\rho_{w}}\left(\frac{\alpha }{c_{p}}\right)\left(\frac{p_{g}^{*}}{p_{e}}\right)exp\left(-\frac{E_{AS}}{RT_{w}}\right)$$
(6)vm⊥=1ρf(Jf0kfcf)12(6p∑σf⊥)32(RTwEAf)12exp(−EAftRTw),vm‖=vm⊥(σf⊥σf)12
where *v_t_* is the total linear rate, *v_o_* is the oxidation rate, *v_s_* is the sublimation rate, *v_m_* is the mechanical erosion rate,$\rho_{w}$ is the density of the laminated CFRP, *M_i_* is the average molecular mass of the substance *i* involved in the oxidation reaction, $M_{o_{2}}$ is the molecular mass of oxygen, $p_{o_{2}w}$ is the partial pressure of oxygen in the airflow, *A_i_* is the pre-exponential factor of oxidation reaction, $\alpha /c_{p}$ is the heat transfer coefficient, $p_{g}^{*}$ is the gas pressure constant, $p_{e}$ is the local pressure, $p_{\sum }$ is the pressure head of airflow on the ablation surface, $\sigma_{f}$ and $\sigma_{f⊥}$ denotes the strength of mono-fiber in the longitudinal and perpendicular directions, respectively. 

It is necessary to consider processes of heat and mass transfer occurring on the surface. The net conduction heat flux can be obtained from thermal equilibrium in the ablation boundary:(7)$$-\lambda \frac{\partial T_{w}}{\partial n}=\alpha q_{\mathrm{laser}}-q_{\mathrm{rad}}-q_{\mathrm{conv}}+q_{\mathrm{oxi}}-q_{\deg}-q_{\mathrm{phas}}$$
where α is the absorption coefficient, $q_{\mathrm{laser}}$ is the heat flux of the laser, $q_{\mathrm{rad}}$, $q_{\mathrm{conv}}$, $q_{\mathrm{oxi}}$, $q_{\deg}$ and $q_{\mathrm{phas}}$ are the heat flux due to surface radiation, convection between composites and airflow, oxidation of residual carbon and fiber, decomposition and pyrolysis of epoxy resin, and phase change and sublimation, respectively. Additionally, $q_{\mathrm{rad}}=\varepsilon \sigma ({T_{w}}^{4}-{T_{0}}^{4})$ where ε is the emissivity of the ablation surface, σ is a Stefan–Boltzmann constant and $T_{0}$ is the environment airflow temperature. $q_{\mathrm{conv}}=h_{w}(T_{w}-T_{0})$ where $h_{w}$ is the coefficient of convective heat transfer. $q_{\deg}=Q_{0}\frac{\partial m}{\partial t}-V\left(\varphi_{g}\rho_{g}{\vec{\upsilon }}_{g}\right)\cdot \nabla h_{g}$ where $Q_{0}$ is the endothermic heat of degradation from the unit mass of the composite. The second part is the convective heat taken away by pyrolysis gas. *V* is the volume of the composite,$h_{g}$ is the enthalpy of the pyrolysis gas, $\rho_{g}$ is the density of the pyrolysis gas, ${\vec{\upsilon }}_{g}$ is the velocity vector of the gas flow, $c_{g}$ is the specific heat of the pyrolysis gas. $q_{\mathrm{oxi}}=C_{bef}\Delta H_{bef}-C_{aft}\Delta H_{aft}$ where *C_i_* is the mass concentration of the component, $\Delta H_{i}$ is the enthalpy of components. $q_{\mathrm{phas}}=\sum m_{i}\int_{T_{-s}}^{T_{s}}c_{i}dT$ where *m_i_* is the mass of components occurring phase changes, and $c_{i}$ is the specific heat of corresponding components. In this paper, the phase change processes of the pyrolytic residual char and carbon fibers are mainly considered.

The deformed meshes are adjusted by Radial Basis Function (RBF) in Step 3, then the Remapping Solution Technology (RST) variables from the old mesh to the new mesh with a process called advection. The principle is that the actual spatial distribution of the material properties and state variables in the old mesh is mapped to the new mesh that has been adjusted for recession. It can deal with the mesh is not allowed to flow from one material domain to another when the material is multilayer. The flowchart of the adaptive mesh algorithm for multilayer material can be shown in Figure 4. The iterative partitioned solution method must solve each single field problem several times until the equilibrium condition is satisfied before moving on to the next time step calculation. The conservation and continuity of all physical quantities transferred at the fluid–solid interface (FSI), including temperature, heat flow, and deformation, are strictly guaranteed.

Then, the numerical modeling is introduced in the following. The three-dimensional parameter of the solid numerical model is 50 × 50 × 2.40 mm. It is modeled according to the ply pattern of the experimental specimen [45°/0°/−45°/90°]_2S_, which contains 16 layers in total, and the thickness of each layer is 0.15 mm. C3D8 8-node linear hexahedron element is used. The number of nodes is 313,638, and the number of elements is 259,200. The fluid numerical model has dimensions of 150 × 120 × 50 mm. It has meshed with a structural grid of 150 × 81 × 152 nodes. The grid near the wall is refined to 0.01 mm in the normal direction and 0.2 mm in the streamlined direction. The boundary conditions are established and named inlet, far field, outlet, no-slip wall, and symmetry.

## 4. Results and Discussions

### 4.1. Experimental Results and Parameters’ Influence

Figure 5 exhibits the instantaneous ablation morphology of the laser irradiation surface when the laser power density is 1273 W/cm^2^ and the Mach number is 3.0. The real-time distributions of carbon fiber can be clearly seen in Figure 5, such as the ply angel being 45°, 0°, −45°, and 90°, respectively. The experimental results demonstrate vast potential for using this in situ measuring technique in extreme thermal–mechanical conditions. On the one hand, the ablation process of material can be clarified. On the other hand, real-time ablation information can be obtained based on the post-process of the experimental data.

Figure 6 shows the comparisons of laser ablation morphology at different conditions. Meanwhile, Figure 7 gives the instantaneous surface recession depth curve that is calculated by using PIVlab results when the velocity direction is transformed, which helps explain the process of real-time ablation in more detail.

Figure 6a depicts the real-time laser ablation morphology of laminated CFRP in a static air environment with a power density of 1273 W/cm^2^. The pyrolysis of epoxy resin matrix on the laser irradiation surface occurs immediately after laser irradiation, resulting in the formation of soot, including the residual carbon. Typically, it includes those progress from cured resin to dehydrated resin to decomposition and char oxidation to leave bare carbon fiber [43,44,45]. However, the soot is rapidly produced when the laser irradiation time is only 0.1 ms. This phenomenon illustrates that the transformation of the first progress is quick, so the dehydrated stage is ignored in this work. The heat-affected zone (HAZ) is elliptical due to its anisotropic thermophysical properties. The central area becomes bright, and the HAZ reverts to a circular shape because of the ever-increasing temperature. The direction of the fibers does not change throughout the laser ablation process, indicating that this layer is yet ablated-through, and the fibers do not undergo oxidation or sublimation reaction.

Figure 6b shows the real-time ablation process of CFRP composites at Mach 1.8 with a power density of 1273 W/cm^2^. The tangential supersonic airflow changes the ablation behaviors of the laser irradiation region. HAZ is not an apparent ellipse. Convective cooling of tangential supersonic airflow is primarily responsible for this phenomenon. Furthermore, as shown in Figure 7, the ply angle of fibers on the laser irradiation surface shifts from 45° to 0° after 1.5 s of laser irradiation. The transition from 0° to −45° ply angle lasts 3.1 s.

Figure 6c illustrates the ablation process under the tangential airflow velocity of Mach 3.0 with a power density of 1273 W/cm^2^. According to Figure 6c and Figure 7, the laminate with a ply angle of 45° (see the fiber direction) lasts up to 1.4 s. The above results demonstrate that the increase in the airflow velocity does not significantly improve mechanical erosion efficiency, much attributed to the low-temperature field in the early stage of laser irradiation. Then the laminate with a ply angle of 0° lasts for 0.9 s (2.3 s in total), the −45° laminate lasts for 1.2 s (3.5 s in total), and the fibers transform to 90°. It should be noted that the final laser ablation depth (LAD) should be corrected by post-test measurement since the last frame of the velocity field has no reference. The above findings show that the TME effect intensifies with the increased temperature, proving that mechanical erosion is a thermal–mechanical coupling behavior. As the laser power density rises to 2546 W/cm^2^, the fiber direction rapidly changes, as illustrated in Figure 6d and Figure 7. The ablation-through of the first laminate needs only 0.6 s. At 4.0 s, a total of 8 layers are ablation-through. The ablation depth is at least twice that of 1273 W/cm^2^.

Detailed post-irradiated samples are given in Figure 8. The laser ablation area can be divided into two regions: the coupled ablation zone (CAZ) and downstream affected zone (DAZ). In CAZ, the uppermost ablative character is the laser ablation pit, which is caused by laser irradiation and high-speed airflow. Although the laser does not irradiate DAZ directly, it still forms tail-shape damage. The matrix begins to pyrolyze when the heat-affected zone expands to DAZ. Meanwhile, fibers without the immobilization of the matrix are formed the damaged morphology under tangential airflow. The tail-shape damage degree is also related to the laser power density and the airflow velocity, and we can use the tail-shape angle to characterize it. As can be seen in Figure 8b, the tail-shape angle is 40.2°. When the tangential airflow increases to Ma 3.0, the angle rises to 59.3°. As the laser power density increases to 2546 W/cm^2^, the angle rises to 80.4°.

### 4.2. Numerical Results and Ablation Mechanisms

As illustrated in Figure 9, the evolution of laser ablation morphology obtained by numerical simulation is compared with the experimental results when the laser power density is 2546 W/cm^2^ and the Mach number is 3.0. The material parameters are given in Appendix A—Table A1. The laser absorption of CFRP is 0.9 when the laser wavelength is 1064 nm [46]. In Figure 9b, each color represents a different ply angle. The coupled thermal-fluid-ablation model using RBF and RST methods can effectively solve multilayer elements’ laser ablation morphology evolution. As a result, the numerical calculation of the ablative process corresponds to experimental results.

A Bruker (Karlsruhe, Germany) DektakXT profiling system scans the samples to investigate the ablation pit (Figure 10a). The comparison of experimental and numerical results at the final laser irradiation time is illustrated in Figure 10a. The ablation depths of each numerical case agree with those of experimental results at the laser irradiation center. The numerical results of ablation depth are 0.38 mm, 0.44 mm, and 1.13 mm, and the corresponding experimental results are 0.36 mm, 0.47 mm, and 1.07 mm, with errors of 5.56%, −6.38%, and 5.61%, respectively, for the cases of laser power density 1273 W/cm^2^ and Mach 1.8, laser power density 1273 W/cm^2^ and Mach 3.0, the laser power density is 2546 W/cm^2^ and Mach 3.0. The ablative profiles are wavy in experimental results and become more evident with the increased laser power density. This phenomenon is related to the microstructure and different ablation rates of each component of the CFRP composite. The result exemplifies the complex non-linear local ablation. Meanwhile, scanning profiles are used to modify the final real-time ablation depth obtained with PIVlab. The SRD curves then become complete. Experimental results indicate that laser power density is the most crucial parameter for laser ablation depth in the range of tangential supersonic airflow. Moreover, the numerical results are consistent with the experimental results. In this view, real-time SRD curves can be used to validate the numerical model from the whole ablation process rather than just the final ablation depth.

The ablation depth caused by each ablation mechanism at the center of the ablation pit is shown in Figure 11. When the laser irradiation time (LIT) is 4.0 s, the contribution of the sublimation to the total depth is 54.36 %, mechanical erosion is 36.54%, and oxidation is 9.1% (Figure 11a). However, when the LIT is less than 2.4 s, mechanical erosion dominates total ablation. The primary reason is that the residual char and fiber begin to strip due to the aerodynamic force of tangential supersonic airflow after the matrix has been pyrolyzed and sublimated. Figure 11b shows the ablation rate curves of each ablative mechanism, providing additional information. The temperature is around 3000 K during the initial stage of laser irradiation (LIT is before 1.5 s). The sublimation rate of carbon fiber and pyrolytic residual carbon is slow in this temperature range, but pyrolysis and matrix sublimation are rapid. The fibers are exposed to tangential supersonic airflow due to losing the bonding function of the matrix. The strength of fibers decreases significantly at high temperatures. As such, the TME rate is faster than the other mechanisms. When the temperature rises to 3300 K, the sublimation rate is accelerated.

It is also observed that there is a transition point in Figure 11b. This is caused by the evolution of ablation morphology, which potentially influences the airflow characteristics. The surface change of the ablation pit is a significant factor influencing the thermomechanical erosion rate. It can be treated as a cavity flow problem [47]. Figure 12 depicts the contours and streamlines of flow velocity and static pressure as ablation morphology evolution. As the ablation depth increases, the cavity flow mode shifts from a closed to an open-cavity regime. There is no significant backward-facing and forward-facing step flow with the impingement of the incoming upstream flow on the cavity wall due to the smooth curved geometry of the ablated pit (Figure 12a,f).

When the open cavity flow is formed (Figure 12b,g, with a start time of about 1.5 s), the free stream flow does not directly enter the ablation pit. A shear layer forms between the free stream and the flow inside the pit. This phenomenon results in a transition point. The shear layer extends the length of the pit and makes contact with the aft wall, causing airflow compression and high-intensity static pressure. Because of the high-intensity static pressure, the erosion rates of the backward-facing region become more significant than those of the forward-facing region. Therefore, as illustrated in Figure 12d,e, the vortex inside the ablated pit moves downstream with the laser ablation process.

## 5. Conclusions

The developed experimental measurement and data processing algorithm can effectively characterize real-time laser ablation behaviors and the surface recession depth (SRD) in a high-speed wind tunnel environment. In addition, a coupled thermal-fluid-ablation model, which deals with the ablative deformation of multilayer materials, was developed to obtain the evolution of ablation morphology and examine each ablation mechanism’s contribution to the total ablation and the influence of flow regime on the TME. The main findings of the study are summarized as follows:(1)In the experimental results, the real-time laser ablation behaviors of CFRP composite illustrate that the TME is a thermal–mechanical behavior. It is related to laser power density and tangential airflow velocity.(2)The velocity distribution of carbon fibers was characterized using an open-source PIVlab. The transform of velocity direction obtains the surface recession depth (SRD).(3)A coupled thermal-fluid-ablation model was established to demonstrate the effect of the flow regime on the TME. When the laser power density is 2546 W/cm^2^ and the Mach number is 3.0, TME dominates total ablation before the laser irradiation time is 2.4 s. When the LIT was 4.0 s, sublimation contributed 54.36% to the total depth, thermomechanical erosion contributed 36.54%, and oxidation contributed 9.10%.(4)The numerical results show that TME is related to the temperature and velocity of tangential airflow and the cavity flow mode. When the mode shifts from the closed cavity regime to the open-cavity regime, the transition moment is revealed by the TME curve. Then sublimation plays a dominant role and the ablation rate of TME gradually decreases.

## Figures and Tables

**Figure 1 materials-16-00790-f001:**
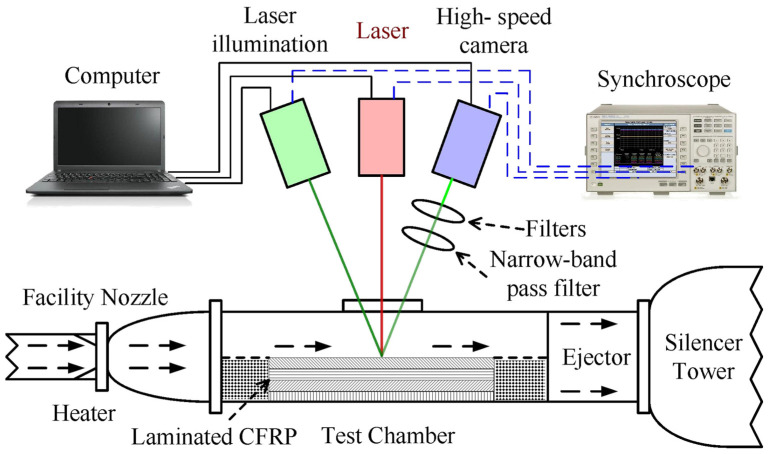
Schematic diagram of experimental setup.

**Figure 2 materials-16-00790-f002:**
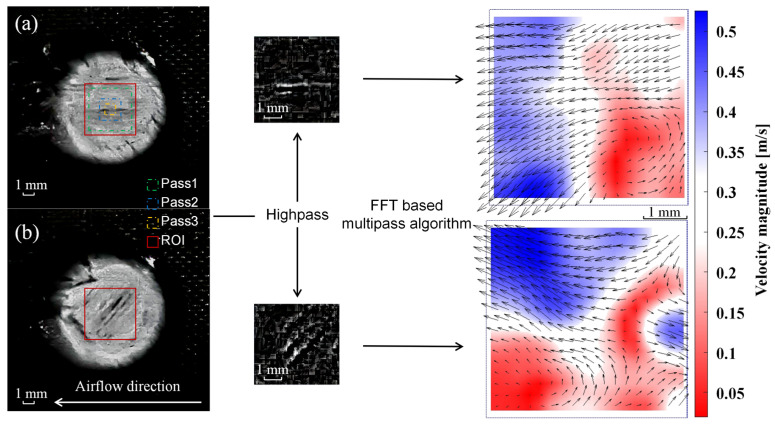
The flowchart of PIVlab analysis and results of velocity distribution. (**a**) Laser irradiation time is 0.8 s; (**b**) Laser irradiation time is 1.2 s.

**Figure 3 materials-16-00790-f003:**
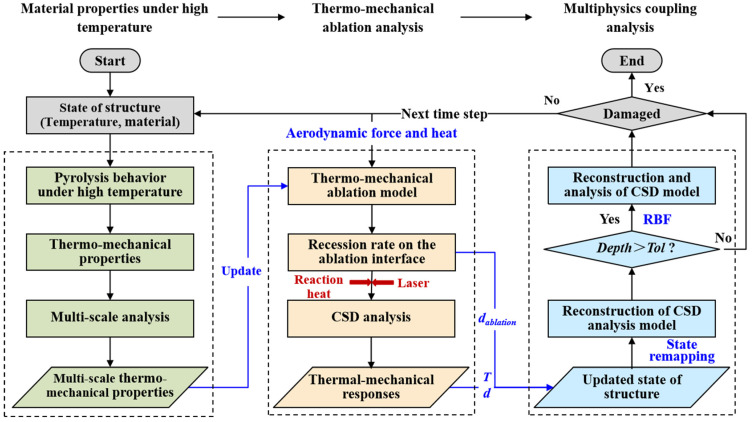
The flowchart of coupled thermal-fluid-ablation analysis.

**Figure 4 materials-16-00790-f004:**

The flowchart of adaptive mesh algorithm for multilayer material.

**Figure 5 materials-16-00790-f005:**
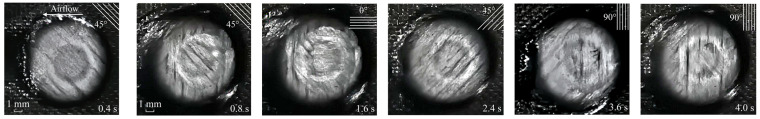
The instantaneous ablation morphology of laser irradiation surface.

**Figure 6 materials-16-00790-f006:**
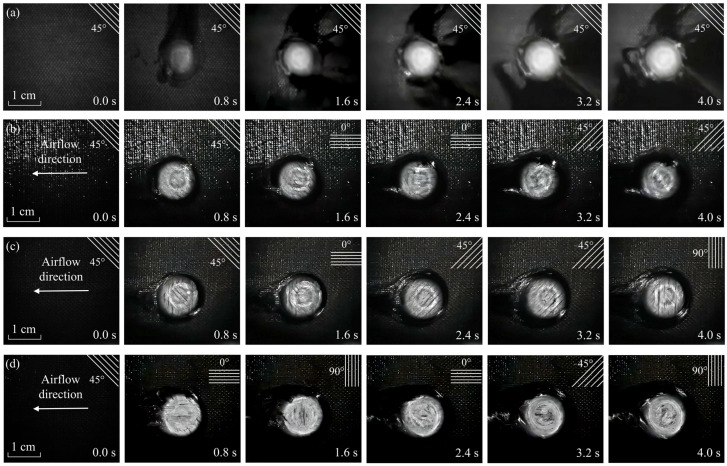
The laser irradiation process in different conditions. (**a**) Static air, 1273 W/cm^2^; (**b**) Ma 1.8, 1273 W/cm^2^; (**c**) Ma 3.0, 1273 W/cm^2^; (**d**) Ma 3.0, 2546 W/cm^2^.

**Figure 7 materials-16-00790-f007:**
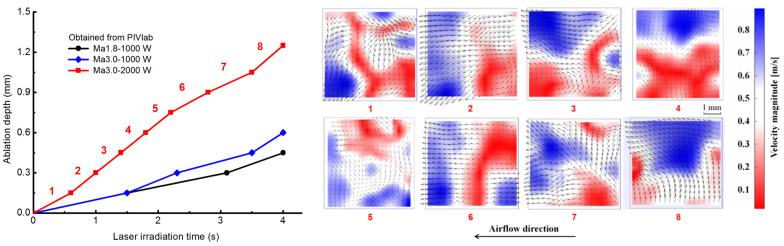
Instantaneous surface recession depths of different experimental conditions are obtained by PIVlab analysis.

**Figure 8 materials-16-00790-f008:**
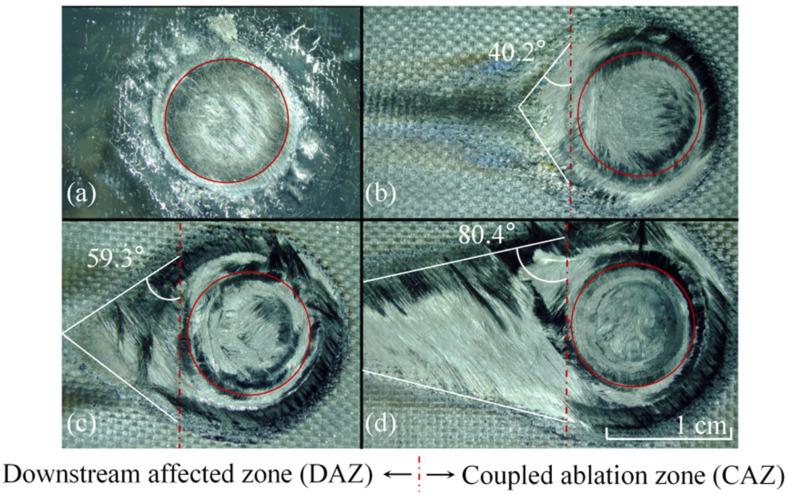
The laser ablation morphology in different condition. (**a**) Static air; (**b**) Ma 1.8, 1273 W/cm^2^; (**c**) Ma 3.0, 1273 W/cm^2^; (**d**) Ma 3.0, 2546 W/cm^2^.

**Figure 9 materials-16-00790-f009:**
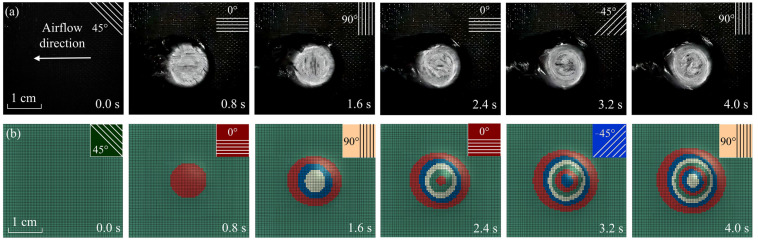
The ablative morphology at different times. (**a**) Experimental results; (**b**) numerical results.

**Figure 10 materials-16-00790-f010:**
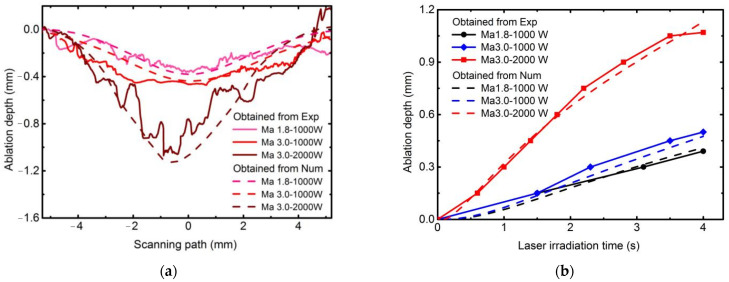
Comparison of experimental and numerical results. (**a**) Profile over the ablation surface; (**b**) instantaneous ablation depth.

**Figure 11 materials-16-00790-f011:**
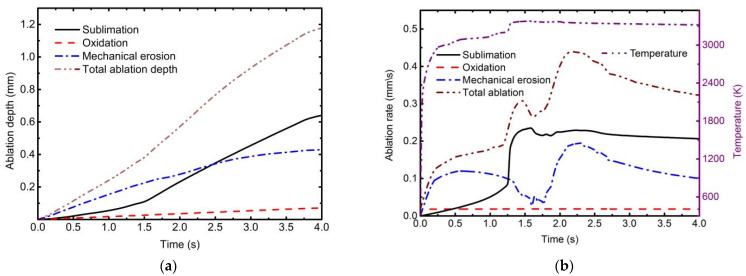
The effect of thermochemical ablation and thermomechanical erosion on laser ablation when laser power density is 2546 W/cm^2^ and Mach number is 3.0. (**a**) Ablation depths; (**b**) ablation rates.

**Figure 12 materials-16-00790-f012:**
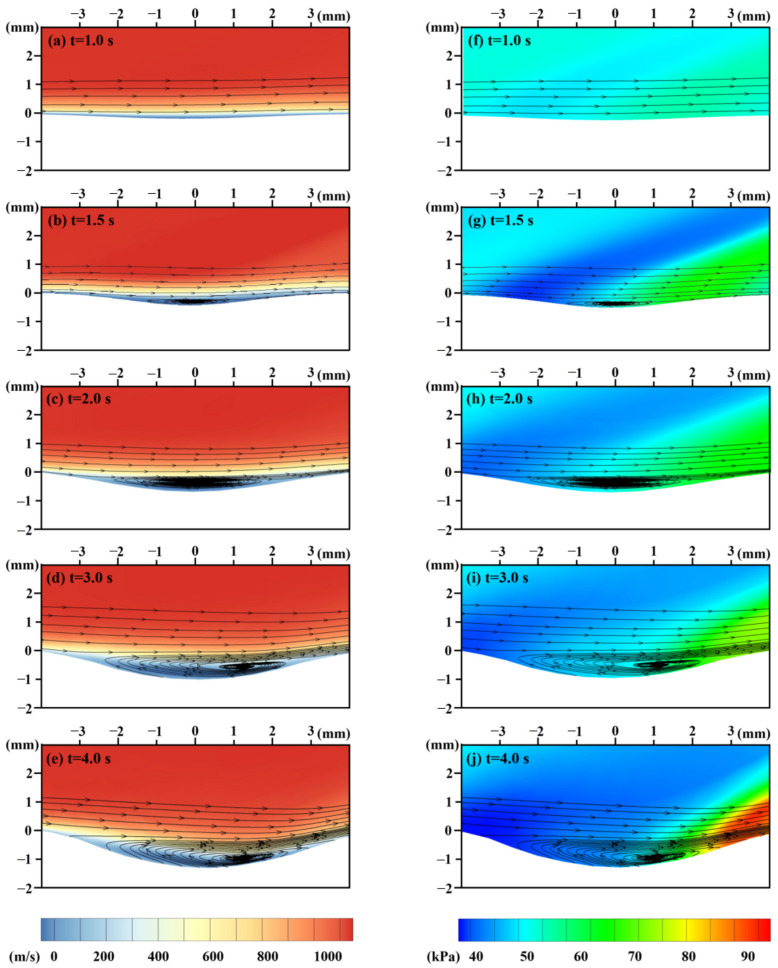
The contours and streamlines of flow velocity and pressure in the fluid domain.

**Table 1 materials-16-00790-t001:** The main physical and mechanical properties of the T700/BA9916 CFRP plates.

Physical Properties (Room Temperature)	Parameter	Mechanical Properties (Room Temperature)	Parameter
Matrix Content (wt%)	38 ± 3	0°Tensile Strength (MPa)	1489
The density of matrix (g/cm^3^)	1.3 ± 0.04	0°Tensile Modulus (GPa)	132.8
The density of fiber (g/cm^3^)	1.78 ± 0.04	90°Tensile Strength (MPa)	58.5
Volatile Content (%)	≤1.5	90°Tensile Modulus (GPa)	9.7
Porosity (%)	≤1.5	Shear Strength (MPa)	121
Lamina Thickness (mm)	0.15 ± 0.015	Shear Modulus (GPa)	5.3

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
