# Peer review of "Instantaneous Ablation Behavior of Laminated CFRP by High-Power Continuous-Wave Laser Irradiation in Supersonic Wind Tunnel"

_materials, 2023, doi:10.3390/ma16020790_

Round 1

Reviewer 1 Report

The authors of the article have presented new results of experimental and computational studies of the processes that develop in the composite material (CFRP) under laser irradiation. The application of new experimental technic and the development of a mathematical model allowed the authors to deepen their understanding of physical/chemical processes during laser ablation of materials which are widely used in aeronautics and aerospace engineering. Therefore, I recommend acceptation of this manuscript after minor revision listed below:

1.      In the introduction, the authors focused on describing the techniques that have already been developed and used by various researchers to study the dynamics of laser ablation. It is desirable to give a more detailed description of the various processes that occur during laser ablation of the CFRP composites and highlight the phenomena that require in-depth study by in situ.

2.      For figures 2 and 7, a scale bar must be added to make the size of the study area clear. It is also desirable to increase the font size for the axis «Velocity….”. For Fig. 7, airflow direction should be indicated.

3.      It would be desirable to clarify whether las   er ablation of the composite resulted in fragmentation/decomposition of all components of the material down to the atomic level, or is it possible to remove particles of micron or smaller sizes from laser-irradiated area?

4.      What factors have impact on the direction of fiber velocity, especially when the velocity vector is against the direction of the airflow? And what effect does the velocity distribution have (or does not have) on the whole ablation process?

Reviewer 2 Report

Dear authors, you did excellent work. Your work offers insightful information on ablation behavior of laminated carbon fiber reinforced polymer (CFRP) exposed to an intense continuous-wave (CW) laser in a supersonic wind tunnel. You chose an interesting and challenging topic to discuss in your paper, which was very well-argued. However, there are still some points required to be covered:

1-     The abstract must be contained to cover the valuable gained results

2-     The instantaneous ablation morphology of laser irradiation surface in figure 5, it was better to take shorter period

3-     The reference in the introduction need to be more recent

4-     The experiment work and used laser parameter were needed to be supported
